# Experience Measures after Radical Prostatectomy: A Register-Based Study Evaluating the Association between Patient-Reported Symptoms and Quality of Information

**DOI:** 10.3390/healthcare10030519

**Published:** 2022-03-12

**Authors:** Ola Christiansen, Øyvind Kirkevold, Ola Bratt, Jūratė Šaltytė Benth, Marit Slaaen

**Affiliations:** 1The Research Centre for Age-Related Functional Decline and Diseases, Innlandet Hospital Trust, 2312 Ottestad, Norway; oyvind.kirkevold@aldringoghelse.no (Ø.K.); j.s.benth@medisin.uio.no (J.Š.B.); marit.slaaen@sykehuset-innlandet.no (M.S.); 2Faculty of Medicine, Institute of Clinical Medicine, University of Oslo, 0450 Oslo, Norway; 3Department of Urology, Innlandet Hospital Trust, 2318 Hamar, Norway; 4Norwegian Advisory Unit, Ageing and Health, 3171 Sem, Norway; 5Faculty of Health, Care and Nursing, NTNU, 2815 Gjøvik, Norway; 6Department of Urology, Institute of Clinical Science, Sahlgrenska Academy, University of Gothenburg, 413 90 Gothenburg, Sweden; ola.bratt@vgregion.se; 7Department of Urology, Sahlgrenska University Hospital, 413 45 Gothenburg, Sweden; 8Health Services Research Unit, Akershus University Hospital, 1478 Nordbyhagan, Norway

**Keywords:** experience measures, robotic-assisted radical prostatectomy, PROMs

## Abstract

Patient-reported data are important for quality assurance and improvement. Our main aim was to investigate the association between patient-reported symptoms among patients undergoing radical prostatectomy and their perceived quality of information before treatment. In this single-centre study, 235 men treated with robotic-assisted radical prostatectomy (RARP) between August 2017 and June 2019, responded to a follow-up questionnaire 20–42 months after surgery. A logistic regression analysis was performed to assess the association between patient-reported symptoms, measured with Expanded Prostate Cancer Index Composite for Clinical Practice (EPIC-CP), and the perceived quality of information. Adverse effects were defined as a higher EPIC score at follow-up than at baseline. The majority (77%) rated the general information as good. Higher EPIC-CP at follow-up was significantly associated with lower perceived quality of information, also after adjustment for age and level of education (bivariate model OR 1.12, 95% CI 1.07; 1.16, *p* < 0.001 and multiple model OR 1.12 95% CI 1.08; 1.17, *p* < 0.001). The share who rated information as good was almost identical among those who reported more symptoms after treatment and those who reported less symptoms (78.3% and 79.2%). Consequently, adverse effects could not explain the results. Our findings suggest a need for improvement of preoperative communication.

## 1. Introduction

According to the World Health Organization, quality health services should be effective, safe, person-centred, timely, equitable, integrated and efficient [1]. Person-centred health care means that individual preferences should be taken into account [2], and implies that assessing user experience is important to secure and improve the quality of care. Patient-reported data capture the patients’ voices, provide information for quality assurance and improvement, and include Patient-Reported Outcome Measures (PROMs), Patient-Reported Experience Measures (PREMs) and patient satisfaction. Examples of PROMs are measures of adverse treatment effects and health-related quality of life (HRQoL). PREMs are defined as person-centred measures evaluating different aspects of interactions with the health care system, such as information and communication [3]. Although the distinction between patient experience and patient satisfaction may sometimes be difficult to capture, PREMs differ from measures of satisfaction [3,4]. PREMs aim to be process indicators that can identify differences in quality of care, for instance, differences in the quality of communication about adverse effects. Satisfaction measures, on the other hand, are subjective and closely related to the patient’s expectation and former experiences [3,4]. 

Prostate cancer is one the most common cancers among men, thus a large number may suffer from adverse effects of their cancer disease and its treatment [5]. Men with localized (non-metastatic) disease have two established, potentially curative treatment options: radiotherapy and surgery [6,7]. Radiotherapy and surgery are equally effective but have different adverse effects [8]. In high-income countries, surgery for prostate cancer is often performed as robotic-assisted radical prostatectomy (RARP). Whereas bowel problems and urinary urgency are the most frequent side-effects after radiotherapy, the most common long-term consequences after RARP are urinary incontinence and erectile dysfunction [9]. All of these adverse effects may affect quality of life [7]. 

When assessing the quality of prostate cancer treatment, the International Consortium for Health Outcomes Measurement recommends that patient-reported adverse effects should be collected with the 26-item Expanded Prostate Cancer Index Composite (EPIC-26) [10,11]. EPIC is available in several different but compatible versions, the shortest of which is the 16-item Expanded Prostate Cancer Index Composite for Clinical Practice (EPIC-CP) [12]. In contrast, there is no similar established questionnaire for PREMs for patients with prostate cancer. Ideally, PREMs should be independent of expectations or outcome [13]. Research suggests that the severity of adverse effects is associated with the grade of satisfaction and regret after radical prostatectomy [14,15], but whether this also applies to experienced quality of information before treatment is scarcely elaborated. 

We have previously tested an adapted version of the PREM questionnaire Quality from the Patients Perspective (QPP) for patients with prostate cancer, which included items concerning patient-perceived quality of information and help to cope with adverse effects [16]. These items were used in the present study with the aim of investigating if the perceived quality of the preoperative information given about RARP and the help received were affected by symptoms after treatment and quality of life. Our primary hypothesis was that men who reported more and severe symptoms on the EPIC-CP rated the quality of the information about adverse effects before treatment as poorer than those who reported better EPIC-CP scores. Our secondary hypothesis was that worse scores on the EPIC-CP sub-domains for urinary incontinence and sexual adverse effects are specifically associated with poorer perceived quality of the information and help given about the related problems. Finally, assuming that the difference between EPIC scores at follow-up and baseline is a measure for adverse effects, we aimed to explore if adverse effects were associated with how patients rated quality of information. 

## 2. Methods

### 2.1. Study Design

The study was a single-centre study based on a local database, developed for quality assurance and research. 

### 2.2. The Database

The TECLA database has previously been described in detail [15]. In addition to clinical and descriptive data such as age and level of education, it includes PROMs (EPIC-CP) and PREMs at baseline and follow-up.

### 2.3. Population

Between August 2017 and June 2019, 361 patients underwent RARP, all of whom were included in the local quality database. Eligible patients for the present study were fluent in Norwegian, had provided informed consent, and had baseline data available, leaving 265 men. Of these, 235 (89%) had filled in a follow-up questionnaire in February 2021, 20 to 42 months after surgery. 

### 2.4. Pre-Operative Information about Adverse Effects and Follow-Up

Once diagnosed, men eligible for radical treatment are discussed in a multidisciplinary meeting consisting of urologists, oncologists and radiologists. Afterwards, they have a pre-treatment consultation with a urologist with experience of RARP and are then informed about treatment options and adverse effects. The information is routinely given both orally and in writing, and the patients are encouraged to use web-based decision aids. If they wish, they are also offered an appointment with a radiation oncologist and/or a peer. During their journey from the pre-treatment consultation to postoperative follow-up, the patients are free to contact a coordinating nurse if they feel insufficiently informed. Patients who report bothersome urinary or sexual problems are offered an appointment with a urotherapist for help with symptom management. 

### 2.5. Assessments

The EPIC-CP contains 16 items, of which 15 cover symptoms from five different domains: urinary incontinence, urinary irritation/obstruction, bowel symptoms, sexual symptoms and vitality/hormonal symptoms. Each domain includes three items scored on a Likert-scale, ranging from 0–4. These item scores are summarized into domain scores ranging from 0–12. The total EPIC-CP score thus ranges from 0 to 60. Higher scores mean more symptoms. 

We have earlier tested an adapted version of the questionnaire Quality from the Patients Perspective (QPP) for collecting PREM, but we could not reproduce QPP’s previously described dimensions [16]. As a result, we no longer routinely use the questionnaire but in this present study, we have included five selected QPP items about communication and coping with adverse effects.

The five retained QPP PREM items cover perceived quality of the information given about adverse effects and the help received to cope with these effects. The items read as follows: “I received good information about adverse effects”, “I received good information about urinary adverse effects”, “I received good information about sexual adverse effects”, “I received help for urinary adverse effects” and “I received help for sexual adverse effects”. These questions were answered on a 4-point Likert scale from “totally agree” (0) to “do not agree at all” (3). “Not applicable” was also an option. For the analyses, the answers were dichotomized into 0–1 (totally agree and largely agree) versus 2–3 (partly agree and do not agree at all). 

### 2.6. Statistical Analysis

Patient characteristics were described as means and minimum and maximum values for continuous variables, and frequencies and percentages for categorical variables. The total EPIC-CP scores, as well as the scores for the urinary incontinence domain (Urinary Incontinence Symptom Score—UISS) and the sexual symptoms domain (Sexual Symptom Score—SSS), were reported as means and standard deviations (SDs) stratified by the dichotomised (totally/largely vs. partially/do not agree at all) answers on the five PREM items. Dichotomization was necessary due to the small category size. Patients with missing answers and “not applicable” were excluded from the analysis. 

To assess the association between EPIC-CP total score and perceived overall quality of the information about adverse effects, a logistic regression analysis was performed. Logistic regression analysis was also performed to assess the associations between EPIC domain scores (urinary incontinence scores and sexual symptom scores) and patients’ perceived quality of information about, and help received for these specific problems. All regression models were adjusted for age and education. We assumed that any association between perceived quality of information and EPIC was represented by an association between perceived quality of information and actual adverse effects (increasing symptoms from baseline to follow-up) or by an association between perceived quality of information and persisting symptoms from baseline. Linear regression analysis with follow-up EPIC score as outcome and baseline EPIC score, how men rated information and interaction between the two as independent variables was performed to assess the differences between those who answered totally/largely agree and those who answered partially/do not agree at all, regarding the association between baseline and follow-up EPIC score. A significant interaction would imply that the interaction between baseline and follow-up EPIC is significantly different between those who answered totally/largely agree and those who answered partially/do not agree at all. Scatter plots were generated for illustrations (Appendix A). Next, differences between baseline and follow-up EPIC scores were calculated and dichotomized to worsening scores or improved scores. χ^2^-test was applied to assess the association between how the information was rated and symptoms. Four men with stable EPIC scores were exclude from this analysis. 

Statistical analyses were performed with SPSS v27. Significance level was set at 5%. 

## 3. Results

Mean age of the study population was 66 years (37–79). In total, 27% had 9 years of obligatory school, 39% had a high school education, and 34% an academic education.

The mean total EPIC-CP was 10.8 (SD 7.8) at baseline and 16.5 (SD 9.6) at follow-up. The mean UISS was 1.0 (SD 1.6) at baseline and 2.7 (SD 2.8) at follow-up. The mean SSS was 3.5 (SD 2.9) at baseline and 7.7 (SD 3.4) at follow-up. 

Of the 235 responders, 182 (77%) totally or largely agreed that they had received good information about adverse effects in general, 178 (76%) totally or largely agreed that they had received good information about urinary incontinence symptoms and 167 (71%) totally or largely agreed that they had received help with such symptoms. Although a majority gave a correspondingly positive answer when rating information on sexual symptoms and help with these adverse effects, fewer men totally or largely agreed on these items: 156 (66%) and 128 (54%) (Table 1).

The mean total EPIC-CP score at follow-up was 16.5 (SD 9.6). For men that answered that they totally or largely agree on the item “I received good information about adverse effects”, the mean EPIC-CP score was 14.3 (SD 8.3), while men that answered that they partially agree or do not agree at all had a mean EPIC-CP score of 24.3 (SD 10.2). The same pattern was found for symptom scores for the urinary incontinence domain and the sexual symptoms domain (Table 2).

In bivariate logistic regression analysis, how men rated the quality of the information they had received about adverse effects was significantly associated with the total EPIC-CP score; this association remained statistically significant after adjusting for age and level of education (Table 3). Significant associations were also found for the perceived quality of information about urinary adverse effects, sexual adverse effects, help to cope with adverse effects and the EPIC-CP scores on the corresponding domains (Table 3). 

More patient-reported symptoms at baseline were associated with more symptoms at follow-up (Appendix A). This association was not statistically significant between those who rated the information as good and those who rated the information as less good (non-significant interaction terms). Of 177 men with all baseline and follow-up data available, 48 reported lower EPIC scores at follow-up compared to baseline, and 129 reported higher EPIC scores at follow-up compared to baseline. There were no differences in how information was rated between men with increase and decrease in symptoms from baseline to follow-up. (Table 4). Among those who reported less symptoms, 79.2% answered that they totally or largely agree on the item “I received good information about adverse effects”, while this was found among 78.3% for men who reported more symptoms (*p* = 0.900 for χ^2^-test).

## 4. Discussion

In this register-based study including RARP patients, we found that although the majority reported having received good information and help with adverse effects, a substantial proportion disagreed or only partly agreed with such statements. These patients also reported more symptoms on follow-up, and the association between higher symptom score and quality ratings was significant and independent of age and level of education. Higher baseline EPIC scores were associated with higher EPIC scores at follow-up. This association did not differ statistically between those who rated the information as good and those who rated the information as less good. 

The association between patient-reported symptoms and perceived quality of information, as demonstrated in this study, has to our knowledge not been reported for radical prostatectomy patients. Our primary hypothesis was confirmed, but our results were not explained by adverse effects as the share of men who reported less symptoms after treatment and rated the quality of information as good was identical to the share who reported more symptoms and rated the quality of information as good. 

Our results are not in line with previous studies addressing patient-perceived quality and outcomes in other surgical settings. In a study by Saarinen et al., more postoperative complications after general and orthopaedic surgery were associated with lower perceived quality of care [17]. Black et al. found a positive association between patient experience and patient-reported outcomes after hip or knee replacement or groin hernia repair [18]. In the large-scale study by Black et al., communication was one of the two aspects of experience that was most related to better outcome (the other was trust in the doctor). Differences between patient groups in the present study and the before mentioned studies could contribute to our results. Orthopaedic patients expect less symptoms after treatment, while surgical prostate cancer patients do not. 

Although previous research on experience measures and outcomes after prostatectomy are scarce, there are studies on satisfaction and treatment regret. A long-term follow-up study found that 15% of surgical prostate cancer patients reported treatment regret, and that regret was more common among men that experienced adverse effects [14]. Other researchers reported that erectile dysfunction after prostatectomy was associated with less satisfaction, but also that improved patient education and more information could improve satisfaction [15].

The distinction between experience and satisfaction is difficult. Although PREMs aim to be independent measures, they are, just as satisfaction measures, influenced by outcomes and expectations. Our findings suggest that not only outcomes, but patient-reported symptoms per se have impact on patient-reported experience. 

The perceived quality of preoperative information is a structure measure that could identify areas to improve. Despite that the majority rated the information as good in our cohort, a notable share rated the given information as less good. Surprisingly, this was also present for men who reported less symptoms after treatment. There are several possible reasons for this. One explanation could be related to timing of the information. Understandably, the focus for many patients recently diagnosed with cancer is to be cured. The prostate cancer diagnosis is for many men a psychological burden [19], so when they meet their surgeon for planning of treatment, they may not be responsive to information about long-term problems. Consequently, they therefore report the quality of the information as poor if adverse effects emerge later on. This may also apply for men who experienced severe symptoms before treatment and later on reported that their symptoms remained or escalated. Another plausible explanation is that surgeons tend to downplay the risk of adverse effects and their severity to avoid worries or are oblivious to how severe the patients actually perceive their problems. There are several studies showing that clinicians underestimate the severity of their patients’ adverse effects [20,21,22,23]. This explanation may also apply to the association between worse EPIC scores and poorer ratings of the quality of help received. A third explanation for our findings could be that the surgeons’ communication skills were not good enough [24]. They may not have been fully able to capture when patients need help, or to present adverse effects in a way that was understood by the patient. A fourth possible explanation is that men with certain personality traits are more likely to report more severe adverse effects [25] and that these men are also more critical against the information they receive. 

Clinical consequences of our results could be to improve urologists’ communications skills and the support to men who experience urinary problems, sexual difficulties or other problems related to prostate cancer and its treatment. Implementation of training programs could be helpful to make urologists attentive on how they communicate and give objective information [26]. It has previously been reported that urologists’ communications skills influence prostate cancer patients’ treatment choices [27]. Men who experience severe new or remaining problems after treatment should further be encouraged to seek help, and during follow-up the clinicians should be aware of their responsibility to offer help to cope with symptom distress and adverse effects. 

A limitation of this study is the single institution design and small sample size. Our results need to be reproduced in multi-institutional and larger scale studies to be generalizable. We also lack information about non-responders. Another limitation is the use of single items and not a validated PREM questionnaire. The selected items have focus on information about and help with adverse effect, hence other aspects of patient-perceived quality of care are not assessed. However, there is no consensus about which PREM questionnaire one should use for prostate cancer patients. In general, PREMs as well as PROMs are often lacking in clinical registries [28], and compared to questionnaires designed for PROMs, there are few validated questionnaires to evaluate PREMs [13]. Another limitation could be recall bias, since the follow-up questions were answered several months after treatment. 

## 5. Conclusions

Patients’ perception of the information and the help they received about adverse effects after radical prostatectomy was associated with self-reported symptoms: more symptoms were associated with poorer patient-perceived quality of information. Adverse effects did not explain this finding. Most men who reported rated the information as good, regardless of whether they had more or less symptoms after than before RARP. Our findings suggest a need for improvement on preoperative communication before RARP. 

## Figures and Tables

**Table 1 healthcare-10-00519-t001:** Descriptive statistics with distributions of answers of perceived quality of information and help to cope with adverse effects at follow-up.

	Totally Agree (0) or Largely Agree (1)	Partially Agree (2) and Do Not Agree at All (3)	Missing or Not Applicable
	N (%)	N (%)	N (%)
I received good information about adverse effects	182 (77)	50 (21)	3 (1.3)
I received good information about urinary adverse effects	178 (76)	52 (22)	5 (2.1)
I received good information about sexual adverse effects	156 (66)	69 (29)	10 (4.3)
I received help for urinary adverse effects	167 (71)	43 (18)	25 (11)
I received help for sexual adverse effects	128 (54)	75 (32)	32 (14)

**Table 2 healthcare-10-00519-t002:** Descriptive statistics of EPIC-CP total score, urinary incontinence symptom score (UISS) and sexual symptom score (SSS) stratified on PREM questions.

Outcome	Totally Agree (0) and Largely Agree (1)	Partially Agree (2) and Do Not Agree at All (3)
	I Received Good Information about Adverse Effects	
EPIC-CP (N = 216)		
N	171	45
Mean (SD)	14.3 (8.3)	24.3 (10.2)
	I received good information about urinary adverse effects	
UISS (N = 229)		
N	178	51
Mean (SD)	2.2 (2.3)	4.1 (3.0)
	I received help for urinary adverse effects	
UISS (N = 224)		
N	156	68
Mean (SD)	2.1 (2.3)	3.4 (3.0)
	I received good information about sexual adverse effects	
SSS (N = 203)		
N	163	40
Mean (SD)	7.6 (3.3)	9.1 (3.2)
	I received help for sexual adverse effects	
SSS (N = 196)		
N	125	71
Mean (SD)	7.1 (3.3)	9.0 (2.7)

EPIC-CP = Expanded Prostate Index Composite for Clinical Practice, UISS = Urinary Incontinence Symptom Score, SSS = Sexual Symptom Score.

**Table 3 healthcare-10-00519-t003:** Results of logistic regression analysis with dichotomized PREM items as outcome.

	Unadjusted Models		Adjusted Model	
	OR (95% CI)	*p*-Value	OR (95% CI)	*p*-Value
I Received Good Information about Adverse Effects (N = 208)				
EPIC-CP at follow-up	1.12 (1.07; 1.16)	<0.001	1.12 (1.08; 1.17)	<0.001
Age			0.96 (0.91; 1.01)	0.144
Level of education				
Obligatory—ref.			1	
High school			0.95 (0.35; 2.57)	0.921
Academic			1.12 (0.47; 2.70)	0.723
I received good information about urinary adverse effects (N = 220)				
UISS	1.27 (1.13; 1.43)	<0.001	1.27 (1.13; 1.43)	<0.001
Age			1.00 (0.95; 1.05)	0.949
Level of education				
Obligatory—ref.			1	
High school			0.68 (0.28; 1.66)	0.393
Academic degree			0.90 (0.42; 1.92)	0.787
I received help for urinary adverse effects (N = 217)				
UISS	1.20 (1.08; 1.34)	<0.001	1.20 (1.08; 1.35)	0.001
Age			0.99 (0.95; 1.04)	0.766
Level of education				
Obligatory—ref.			1	
High school			0.92 (0.41; 2.04)	0.835
Academic			1.20 (0.60; 2.39)	0.607
I received good information about sexual adverse effects (N = 195)				
SSS	1.16 (1.03; 1.30)	0.018	1.19 (1.05; 1.34)	0.007
Age			0.98 (0.92; 1.04)	0.473
Level of education				
Obligatory—ref.			1	
High school			0.39 (0.14; 1.07)	0.068
Academic			0.76 (0.34; 1.72)	0.510
I received help for sexual adverse effects (N = 189)				
SSS	1.23 (1.11; 1.36)	<0.001	1.27 (1.13; 1.43)	<0.001
Age			0.99 (0.94; 1.05)	0.837
Level of education				
Obligatory—ref.			1	
High school			0.28 (0.11; 0.67)	0.005
Academic			0.72 (0.35; 1.51)	0.385

**Table 4 healthcare-10-00519-t004:** Descriptive statistics on the item “I received good information about adverse effects” stratified on men who reported less and more symptoms after treatment (defined as the difference between EPIC score at follow-up and baseline).

I Received Good Information about Adverse Effects	Totally Agree (0) and Largely Agree (1)	Partially Agree (2) and Do Not Agree at All (3)	
Total, N (%)	139 (78.5)	38 (21.5)	177
Men with less symptoms, N (%)	38 (79.2)	10 (20.8)	48
Men with more symptoms, N (%)	101 (78.3)	28 (21.7)	129

## Data Availability

The dataset used and analyzed during this study are available from the corresponding author on request.

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
