# Peer review of "Experience Measures after Radical Prostatectomy: A Register-Based Study Evaluating the Association between Patient-Reported Symptoms and Quality of Information"

_healthcare, 2022, doi:10.3390/healthcare10030519_

Round 1

Reviewer 1 Report

The authors perform a register-based study evaluating the association between patient-reported symptoms and quality of information. The cohort comprises of robotic-assisted radical prostatectomy with a follow-up questionnaire post surgery approximately 20-42 months. The study looks at expanded prostate cancer index composite for clinical practice (EPIC-CP) and the perceived quality of information. The study concludes that higher EPIC-CPS follow-up was significantly associated with lower perceived quality of information. Additionally noted is that adverse effects could not explain the results. Overall, the authors suggest the need for improvement of preoperative communication.

The overall design of the study is valid and the authors make valid conclusions. Limitations of the study include a small sample size with reporting of a single institution design. The overall results are interesting in that the ability to assess for preoperative communication before radical prostatectomy and aiding the overall perception of adverse effects is an important consideration for clinicians. Overall, the study highlights the importance of preoperative communication through a questionnaire setting. 

Author Response

Thank you very much for taking the time to review our manuscript. 

Reviewer 2 Report

This paper investigates the relationship if any between the self-reported symptoms after the radical prostatectomy treatment and information/help provided regarding the adverse post-treatment symptoms.

In Table 3, maybe "Multivariate model" instead of "Multiple model"? It might also be useful to clarify what bivariate model means here. It might refer to the association between one dependent and one explanatory variable (I believe that's the case here), or two binary dependent variables and one or more explanatory ones. Maybe including the formula which was used at the bottom of the table would be useful?

I would also suggest reporting actual p-values instead of just indicating that they passed a significance threshold. This will contribute to the reproducibility and better understanding of the study. If the p-values are too small, they can be presented in the scientific notation.

Results: I am not quite clear what the study shows at the end. There is a significant association between more self-reported adverse symptoms after the treatment and the perceived lack of information. On the other hand, the authors state that adverse symptoms are irrelevant, because of the similar proportions of patients reporting the lack of information regardless. This sounds contradictory and confusing to me. So, is there a difference between the patients with less and patients with more symptoms reporting the lack of pre-treatment information or not? Is there a significant association between patients with less symptoms and the perceived lack of information too? I think this section needs to be expanded to include more discussion and possibly additional analyses.

Small corrections:

l.10 "..was estimated..", maybe ".. was used.." or ".. was applied.."? Alternatively, the model's parameters were estimated..

l.87 Not sure whether it was meant to sound that men "are discussed" in a meeting, or men discuss the available options with the doctor in a meeting?

l.127 same correction as in l.10

l.121 It is not clear at this point in the paper why the abbreviation for the urinary incontinence domain is UISS instead of UID for example? It is explained further in the paper, but it would be better to mention it here when the abbreviation is first introduced.

Round 2

Reviewer 2 Report

I am generally satisfied with the improvements made to the paper. I have a few outstanding questions so:

1) l.147 - it would be good to add a reference here to the Supplementary Figure (a scatter plot).
2) l.149 - the authors claim they performed the chi-squared test, but I cannot find the results anywhere. If they are mentioned in the Supplementary materials, then the reference here would be useful, but I cannot seem to find them in the supplementary either. I am not sure what the result of this test is either.

As for the p-values, I would still advise the authors to disclose full p-values regardless of how small they are. There is a big difference between 1e-03 and lets say 1e-10 in terms of how confident we are about any result. Although certain p-value thresholds are widely applied and acceptable in the scientific community, they are inevitably arbitrary and often lead to the biased or incomplete understanding of the results. However, I will leave this decision up to the authors.
